# INSTRUCTEDIT: IMPROVING AUTOMATIC MASKS FOR DIFFUSION-BASED IMAGE EDITING WITH USER INSTRUCTIONS

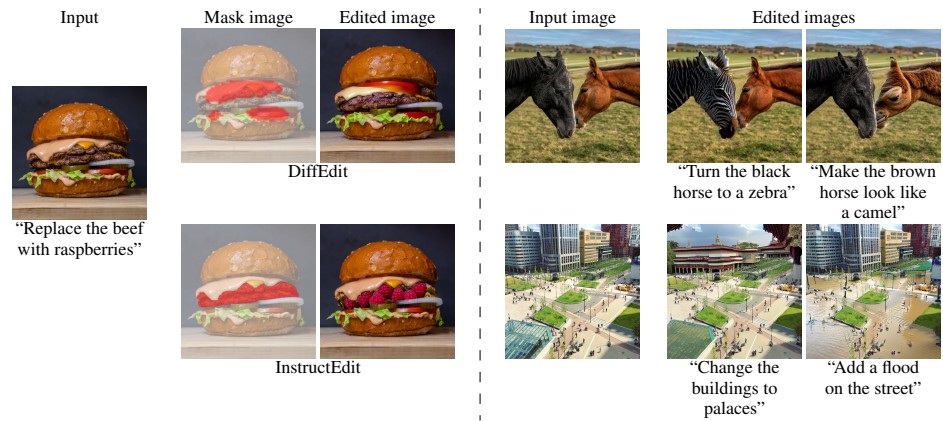

Figure 1: Left: a comparison between DiffEdit and InstructEdit. Right: examples of editing using InstructEdit. Note that InstructEdit only requires user instructions as input, while DiffEdit needs an input caption and an edited caption instead.

## ABSTRACT

Recent works have explored text-guided image editing using diffusion models and generated edited images based on text prompts. However, the models struggle to accurately locate the regions to be edited and faithfully perform precise edits. In this work, we propose a framework termed **InstructEdit** that can do fine-grained editing based on user instructions. Our proposed framework has three components: language processor, segmenter, and image editor. The first component, the language processor, processes the user instruction using a large language model. The goal of this processing is to parse the user instruction and output prompts for the segmenter and captions for the image editor. We adopt ChatGPT and optionally BLIP2 for this step. The second component, the segmenter, uses the segmentation prompt provided by the language processor. We employ a state-of-the-art segmentation framework Grounded Segment Anything to automatically generate a high-quality mask based on the segmentation prompt. The third component, the image editor, uses the captions from the language processor and the masks from the segmenter to compute the edited image. We adopt Stable Diffusion and the mask-guided generation from DiffEdit for this purpose. Experiments show that our method outperforms previous editing methods in fine-grained editing applications where the input image contains a complex object or multiple objects. We improve the mask quality over DiffEdit and thus improve the quality of edited images. We show that our framework can accept multiple forms of user instructions as input.

## 1 INTRODUCTION

Generative diffusion models are a versatile tool to generate images (Saharia et al., 2022; Ramesh et al., 2022; Balaji et al., 2022; Rombach et al., 2022; Li et al., 2023b), videos (Ho et al., 2022b;

Esser et al., 2023; Ho et al., 2022a; Blattmann et al., 2023; Yang et al., 2022), and 3D shapes (Hui et al., 2022; Zhang et al., 2023; Gupta et al., 2023; Zheng et al., 2023). In addition, the powerful representation learned by generative diffusion models makes them a great basis for downstream editing operations. In this paper, we are particularly interested in image editing operations.

Training-free and tuning-free text-guided image editing using diffusion models (Hertz et al., 2022; Couairon et al., 2022; Wang et al., 2023; Liew et al., 2022) usually relies on descriptive text captions for both input image and edited image. However, one line of work focuses on accepting human instructions as input to edit images, as human instructions are more intuitive for a user to provide, and can be free from specific prompting structures. InstructPix2Pix (Brooks et al., 2022) constructs an "input caption, edited caption, user instruction" triplet dataset by fine-tuning GPT-3 (Brown et al., 2020) and generates pairs of input image and corresponding edited image using Prompt-to-Prompt (Hertz et al., 2022). Here we also try to accept human instructions as input. Instead of fine-tuning a large language model, we utilize its in-context learning ability to parse user instructions on the fly. One challenge of this tuning-free approach is that the user instructions can be as unclear as "Add glasses" or "Turn him into a bearded man", where no information about the referred object is given. To tackle this problem, we utilize a large language model to understand the user instructions along with a multi-modal model to improve comprehension.

Using a mask to provide guidance to the models about the specific areas to edit is a logical and intuitive approach. There already exist several editing models (Brooks et al., 2022; Hertz et al., 2022; Liew et al., 2022; Wang et al., 2023) that do not require a mask as input. This approach generally works well when there is a single object in the image or if there is no object of the same type as the object the user wants to edit. However, with multiple objects in the input image, diffusion models struggle to correctly identify which object the user intends to edit. For example, if the user wants to edit a specific object like "A yellow chair" in an image with multiple chairs present or "The cat on the left" in a group of cats. In such cases, an input mask can help the model to correctly locate the object of interest. As InstructPix2Pix utilizes the mask-free image editing method Prompt-to-Prompt to generate the paired input-edited images, its ability to accurately locate objects is not sufficient in complex cases. One possible solution is to create more paired images and paired captions to fine-tune InstructPix2Pix. Here we provide another solution by adopting a pre-trained image segmentation model for generating high-quality masks to guide the editing. This does not require any training or dataset collection.

While it is possible to ask a user to paint a mask, it requires time-consuming and detailed user interaction. We therefore follow the previous work DiffEdit that employs automatically generated masks. While DiffEdit obtains very good masks in many cases, there are several instances where DiffEdit fails to produce high-quality masks: 1) the descriptive captions are not informative enough; 2) the threshold of the mask filtering is not correctly set; 3) there is more than one object of the same or different type; and 4) there are many parts of one object in the image that may cause ambiguity. In our work, we set out to tackle the challenge of computing improved masks in such challenging cases. We adopt a powerful grounding segmentation module called Grounded Segment Anything (Grounded SAM). By simply providing the segmentation prompt to Grounded SAM, a mask that exactly matches the shape of the object can be generated. This provides extra grounding and segmentation ability to the framework, which helps the model to locate and extract the object(s) to be edited.

In this work, we propose a framework called **InstructEdit** to use large pre-trained models to edit images following user instructions based on DiffEdit. We automatically extract higher-quality masks compared to DiffEdit, and therefore achieve more preferable and stable editing results in more complex multi-object image editing scenarios. Specifically, our method has three components: a language processor, a segmenter, and an image editor. We first use the language processor to understand the user instruction by identifying which object(s) should be edited and how. This first step yields the segmentation prompt for the segmenter and captions for the image editor. We use ChatGPT to parse the user instruction and optionally adopt BLIP2 (Li et al., 2023a) when the user instruction is unclear. The segmenter then accepts the segmentation prompt and generates a mask that outlines the region according to the segmentation prompt. We utilize Grounded Segment Anything as the segmenter which combines both high grounding and segmentation abilities. Finally, the image editor performs image editing using the captions along with the generated mask. We adopt Stable Diffusion along with the mask-guided image editing process to do the editing.

We show that our method outperforms previous editing methods in fine-grained editing applications. Specifically, we are interested in input images that contain 1) one object and we want to edit one part of the object. 2) multiple objects and we want to edit one or multiple objects. We show that we improve the quality of edited images by improving the mask quality over DiffEdit. We also show that our framework can accept multiple forms of user instructions as input. We summarize our contributions as below:

- We propose a diffusion-based text-guided image editing framework that accepts user instruction as input instead of input caption and edited caption.
- We outperform baseline methods in fine-grained editing when we want to edit one part of the object in a single-object image or one or multiple objects in a multi-object image.
- We improve the mask quality over the original DiffEdit and thus improve the image quality.
- Our framework can be combined with the NeRF editing pipeline to achieve fine-grained scale NeRF editing application.

## 2 RELATED WORK

### 2.1 IMAGE EDITING USING DIFFUSION MODELS

Pre-trained diffusion models (Ramesh et al., 2022; Saharia et al., 2022; Rombach et al., 2022; Huang et al., 2023a) can be used to do various image editing tasks. Several works (Valevski et al., 2022; Kawar et al., 2022; Kwon & Ye, 2022) fine-tune the diffusion model weights or optimize a loss function to perform image editing. However, in these works fine-tuning requires a relatively long time to obtain a single edited image, and each editing prompt requires its own fine-tuning process.

Many works (Meng et al., 2021; Hertz et al., 2022; Parmar et al., 2023; Wang et al., 2023) achieve good image editing results using a tuning-free approach. Prompt-to-Prompt (Hertz et al., 2022) proposes to edit the cross-attention maps by comparing the input caption and the edited caption. MDP (Wang et al., 2023) proposes to manipulate the diffusion path by analyzing the sampling formula. In this line of work, a mask is not required during the editing process, thereby making it easier to use than systems that rely on masking. These kinds of methods usually perform well when there is a single foreground object in the image, but fail when more fine-grained control is needed.

By providing a manually designed mask as input, a user can explicitly control which region to edit and which region to preserve. Blended Diffusion (Avrahami et al., 2022b) and Blended Latent Diffusion (Avrahami et al., 2022a) utilize a mask to perform text-guided image editing by operating either in pixel space or in latent space. Shape-guided Diffusion (Park et al., 2022) proposes to use a mask with inside-outside attention to preserve the shape of the object to be edited. MasaCtrl (Cao et al., 2023) focuses on performing non-rigid editing while using a mask to alleviate the confusion of foreground with background objects.

Although a mask provides additional control in the editing process, generating a manual mask can be a burden for the user if the mask is not automatically estimated. (Park et al., 2022; Cao et al., 2023) can also replace the manual mask with an automatic mask generated by cross-attention maps. Another prominent work DiffEdit (Couairon et al., 2022) proposes a way to automatically predict a mask by contrasting the predicted noise conditioned on the input caption and the edited caption. In this work, we intend to exploit the benefit of using a mask without bringing the burden of manually labeling a mask to the user. We adopt large pre-trained models to help us automatically infer a mask.

Several recent video editing papers that extend the concepts from image editing to videos (Liu et al., 2023a; Wu et al., 2023; Qi et al., 2023; Ma et al., 2023; Molad et al., 2023). While video editing is beyond the scope of our work, it is very interesting for future research, as our proposed editing capabilities are not yet available in video editing frameworks.

### 2.2 FOUNDATIONAL MODELS

**Large language models.** The development of large language models has been a rapidly evolving field in recent years (Radford et al., 2018; 2019; Brown et al., 2020; Devlin et al., 2019; Raffel et al., 2020; Clark et al., 2020). One important contribution is the GPT series of models (Radford et al.,

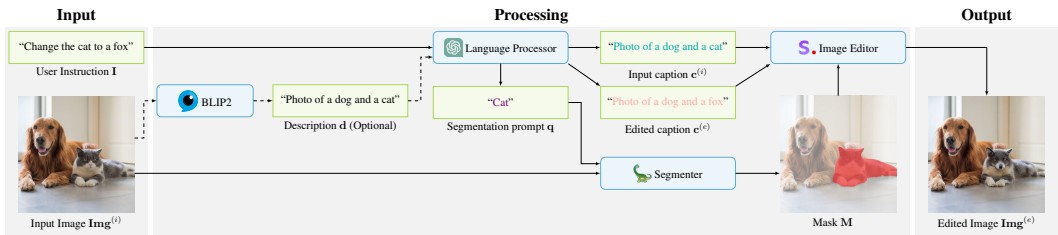

Figure 2: Pipeline: given a user instruction, a language processor first parses the instruction into a segmentation prompt, an input caption, and an edited caption. A segmenter then generates a mask based on the segmentation prompt. The mask along with the input and edited captions are then going to an image editor to produce the final output.

2018; 2019; Brown et al., 2020). These models are pre-trained on massive amounts of text data and then fine-tuned for specific tasks. Most notably, ChatGPT (OpenAI, 2023) is a variant of the GPT series of models that are designed specifically for generating human-like responses in conversational settings. We make use of ChatGPT in our work for the purpose of information extraction from user instruction.

**Segmentation model and grounding detector.** Segment Anything (Kirillov et al., 2023) is a segmentation model that uses a combination of different input prompts and enables zero-shot generalization to unfamiliar objects and images without requiring additional training. Grounding DINO (Liu et al., 2023b) is an open-set object detector that combines the Transformer-based detector DINO (Zhang et al., 2022) with grounded pre-training to detect arbitrary objects with human inputs such as category names or referential expressions. Their system outputs multiple pairs of object boxes and noun phrases give a prompt.

**Images and language.** Vision language models (VLMs) (Radford et al., 2021; Alayrac et al., 2022; Mañas et al., 2022; Huang et al., 2023b; Li et al., 2023a) are a powerful class of models that combine computer vision and natural language processing. VLMs have gained significant attention in recent years due to their ability to bridge the gap between visual and textual information, enabling a range of applications such as image captioning, visual question answering, and image retrieval. Especially, BLIP-2 (Li et al., 2023a) unlocks the capability of zero-shot instructed image-to-text generation. Given an input image, BLIP-2 can generate various natural language responses according to the user's instruction.

## 3 METHOD

### 3.1 PRELIMINARIES

During the training process of diffusion models, we have the objective function

$$\min_{\theta} \mathbb{E}_{\mathbf{x}_0, \boldsymbol{\epsilon}, t} \left\| \boldsymbol{\epsilon} - \boldsymbol{\epsilon}_\theta(\mathbf{x}_t, \mathbf{c}, t) \right\|^2, \tag{1}$$

where $\mathbf{x}_0$ is the input image, $\boldsymbol{\epsilon} \sim \mathcal{N}(\mathbf{0}, \mathcal{I})$ is Gaussian noise that is added to the input image and $\boldsymbol{\epsilon}_\theta$ is the noise estimator that is used to predict the added noise. $t$ is the denoising timestep while $\mathbf{c}$ is the condition of the diffusion model. In this work, we only consider $\mathbf{c}$ to be a text prompt of a text-guided diffusion model. After training, the noise estimator $\boldsymbol{\epsilon}_\theta$ can be used to generate new samples. We use DDIM (Song et al., 2021), which is a deterministic sampler with denoising steps

$$\mathbf{x}_{t-1} = \sqrt{\alpha_{t-1}} \cdot f_\theta(\mathbf{x}_t, \mathbf{c}, t) + \sqrt{1 - \alpha_{t-1}} \cdot \boldsymbol{\epsilon}_\theta(\mathbf{x}_t, \mathbf{c}, t), \tag{2}$$

where $f_\theta(\mathbf{x}_t, \mathbf{c}, t) = \frac{\mathbf{x}_t - \sqrt{1-\alpha_t} \cdot \boldsymbol{\epsilon}_\theta(\mathbf{x}_t, \mathbf{c}, t)}{\sqrt{\alpha_t}}$, and $\alpha_t$ is the noise schedule factor in DDIM. We denote $\boldsymbol{\epsilon}_t = \boldsymbol{\epsilon}_\theta(\mathbf{x}_t, \mathbf{c}, t)$. Given an input image, we can use DDIM inversion to invert it into an initial noise tensor $\mathbf{x}_T$. Each inversion step is calculated as

$$\mathbf{x}_{t+1} = \sqrt{\alpha_{t+1}} \cdot f_\theta(\mathbf{x}_t, \mathbf{c}, t) + \sqrt{1 - \alpha_{t+1}} \cdot \boldsymbol{\epsilon}_\theta(\mathbf{x}_t, \mathbf{c}, t). \tag{3}$$

We iteratively apply this formula until obtaining $\mathbf{x}_T$. However, if we stop the inversion step at timestep $r \leq T$, we encode $\mathbf{x}_0$ into a less noised version $\mathbf{x}_r$. $r$ is called the encoding ratio as in DiffEdit. A larger value $r$ indicates a stronger editing effect, making the edited image guided more by the edited caption but look less like the input image.

## 3.2 LANGUAGE PROCESSOR

Given a user instruction $\mathbf{I}$ as input, we use a large language model ChatGPT (OpenAI, 2023) to extract segmentation prompt $\mathbf{q}$ for the segmenter, and an input caption $\mathbf{c}^{(i)}$ and an edited caption $\mathbf{c}^{(e)}$ for the image editor. Here, we utilize the in-context learning ability of large language models (Dong et al., 2023) to achieve zero-shot task learning. In-context learning does not require any tuning of the parameters of the language model. Instead, it learns the pattern from the task examples and makes predictions when it sees a new example. In this work, by giving a few examples, ChatGPT is able to learn the task and follow the instructions. We show one task example in the Supplementary Materials. A task example shows how ChatGPT should manipulate a user instruction that is provided as input.

When the user instruction $\mathbf{I}$ or the description of the object to be edited is unclear, it is difficult for ChatGPT to correctly provide the prompts as it has no access to the content of the input image. The vision-language model BLIP2 can process the image and is able to answer questions about its content. BLIP2 can provide a short description of the original image, which can assist ChatGPT in providing prompts for the segmenter and captions for the image editor.

In this work, we optionally query BLIP2 to obtain a description $\mathbf{d}$ of the image. Given an input image, we first ask BLIP2 *"Is this a photo, a painting, or another kind of art?"*. We denote the answer as $\rho$ and reuse it in another query to BLIP2 composed as *"$\rho$ of"* to obtain a completed sentence describing the image as input prompt to ChatGPT. ChatGPT in turn can refine the prompt by identifying which object to edit and provide more details even when the user instruction does not specify the content to be edited or the description is incomplete to unambiguously refer to the intended objects in the image.

## 3.3 SEGMENTER

We use Grounded Segment Anything as our segmenter to locate the object(s) to be edited and to compute a corresponding mask. Grounded Segment Anything is a framework which combines Grounding DINO (Liu et al., 2023b) and Segment Anything (Kirillov et al., 2023). Grounding DINO is an open-set object detector, which can accept a given text and output one or multiple detected bounding boxes and a text similarity score per bounding box. Segment Anything (SAM) is a powerful segmentation model. It can accept the bounding box output by Grounding DINO and produce high-quality binary masks for the downstream tasks.

Grounding DINO is first applied to get a bounding box for a given segmentation prompt $\mathbf{q}$ by

$$\text{DINO}(\mathbf{x}_0, \mathbf{q}) = \mathbf{b} = [h, w, \Delta h, \Delta w],$$

where $[h, w]$ is the top-left corner coordinate of the detected bounding box in pixel space, and $[\Delta h, \Delta w]$ is the size of the bounding box. Then, the bounding box is refined to a per-pixel binary mask $\mathbf{M}$ by

$$\text{SAM}(\mathbf{x}_0, \mathbf{b}) = \mathbf{M}.$$

## 3.4 IMAGE EDITOR

We adopt the mask-guided image editing as in DiffEdit (Couairon et al., 2022). Given an input image $\mathbf{Img}^{(i)}$ (which is also denoted as $\mathbf{x}_0$ in 3.1), we want to edit it to get the edited image $\mathbf{Img}^{(e)}$. With the automatically generated mask $\mathbf{M}$ and an encoded noise $\mathbf{x}_r$, the mask-guided DDIM denoising step is formulated as

$$\widetilde{\mathbf{y}}_t = \mathbf{M}\mathbf{y}_t + (1 - \mathbf{M})\mathbf{x}_t, \tag{4}$$

where $\mathbf{y}_t = \begin{cases} \mathbf{x}_r & \text{if } t = r, \\ \boldsymbol{\epsilon}_\theta(\mathbf{y}_{t-1}, \mathbf{c}, t) & \text{otherwise.} \end{cases}$ The region within the mask will have the changes guided by the edited caption, while the region outside the mask will be mapped back to the original pixels. We obtain $\mathbf{Img}^{(e)} = \widetilde{\mathbf{y}}_0$ after iteratively applying Eq. 4.

Table 1: Quantitative comparisons between our method and baselines.

| | MDP-$\epsilon_t$ | InstructPix2Pix | DiffEdit | **InstructEdit** |
|---|---|---|---|---|
| LPIPS ↓ | 0.214 | 0.290 | 0.167 | **0.121** |
| CLIP score ↑ | 26.414 | 25.844 | 26.847 | **27.404** |
| CLIP directional similarity ↑ | 0.079 | **0.114** | 0.106 | 0.082 |

Figure 3: Comparison of the masks (colored in red and blended with the input image) and the corresponding edited image (below each mask) generated by DiffEdit and InstructEdit.

## 4 EXPERIMENTS

### 4.1 SETTINGS

**Baselines.** We choose three text-guided image editing methods that do not require a manual mask as input as our baselines. MDP-$\epsilon_t$ (Wang et al., 2023) is a mask-free editing method that manipulates the diffusion paths by mixing the predicted noise guided by the input caption and the edited caption according to a defined mixing schedule. InstructPix2Pix (Brooks et al., 2022) is also a mask-free editing method. It trains a conditional diffusion model that accepts both an image and a user instruction as input to edit the input image. It highlights the user instruction as input without edited captions by exploiting a large language model. DiffEdit (Couairon et al., 2022) is a mask-based editing method that produces an automatically computed mask by subtracting the predicted noises guided by the input caption and the edited caption. InstructEdit adopts the same mask guidance as in DiffEdit to generate the edited image. However, InstructEdit uses a pre-trained segmentation model to automatically compute a new mask.

All the experiments are performed on a single NVIDIA A100. We use Stable Diffusion v1.5 as the backbone of the image editor. We use the model's weights and implementation of Grounded Segment Anything from [1]. More details can be found in the Supplementary Materials.

**Evaluation.** We provide both qualitative and quantitative results for our method. For quantitative metrics, we use LPIPS (Zhang et al., 2018) to measure the similarity between the input image and the edited image, CLIP score (Hessel et al., 2022) to measure the instruction-image compatibility and CLIP directional similarity (Gal et al., 2021) to see if the change in images is consistent with the change in captions. As quantitative metrics alone usually cannot align with human judgment, we additionally conduct a user study for a better evaluation.

---

[1]https://github.com/IDEA-Research/Grounded-Segment-Anything

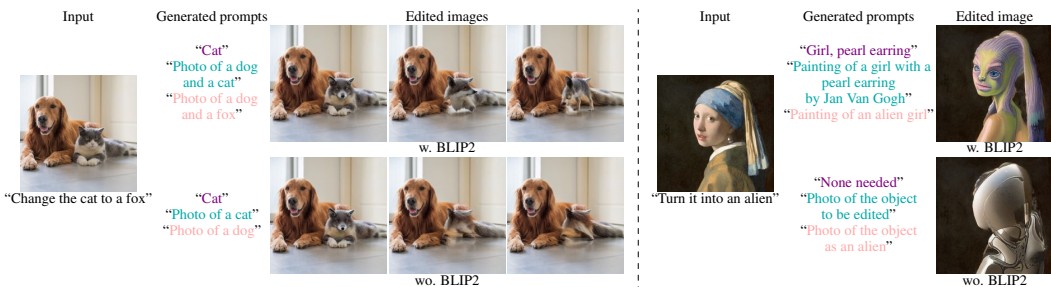

Figure 4: Examples of how BLIP2 improves the quality of edited images by improving the generated prompts. The three generated prompts are segmentation prompt, input caption, and edited caption, respectively. For the first example, the BLIP2 description of the image is "Photo of a dog and a cat". We show three examples for w./wo. BLIP2 with increasing encoding ratio $r$ from left to right. In the second example, the BLIP2 description is "Painting of a girl with a pearl earring by Jan Van Gogh". (BLIP2 gets the name of the painting correct but the name of the artist is wrong.)

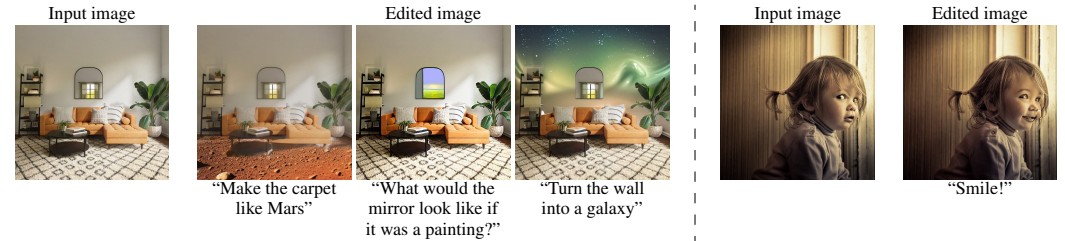

Figure 5: Examples of the variety of the input user instructions. Under each edited image is the input instruction.

## 4.2 BASELINE COMPARISONS

We provide selected qualitative comparisons between our method and other baselines in Fig. 6. It can be seen that InstructEdit can accurately locate either an object or a part of an object to be edited according to the user instruction in a fine-grained manner. In addition, the edits are faithfully performed within the regions InstructEdit identifies. The three baselines have difficulty to accurately locate the object of interest. We show more examples in Sec. 4.3 to show that InstructEdit outperforms DiffEdit by improving the quality of masks.

We show the results of the quantitative evaluation in Tab. 1 for the 10 editing examples shown in Fig. 6. Results show that our method has the best semantic preservation of the edited image compared to the input image and the best alignment of the instruction and edited image pair. The metrics do not capture the large improvements due to our method, because CLIP itself does not capture the fine-grained spatial localization required to judge our complex edits. We therefore perform a user study for these 10 editing examples by comparing our method with the other baseline methods. The results show that InstructEdit was preferred over MDP-$\epsilon_t$, InstructPix2Pix, and DiffEdit in 83.0%, 83.0%, and 84.5% of the cases, respectively.

## 4.3 MASK IMPROVEMENT

We show examples of how InstructEdit improves the mask quality over the original DiffEdit in Fig. 3. For each example, we use the user instruction as input to InstructEdit and design specific input captions and edited captions for DiffEdit. We also select three different mask thresholds $\theta$ for DiffEdit to show how the mask threshold influences the mask quality and therefore the image quality. We show that for DiffEdit the generated mask cannot accurately outline the intended region as specified in the user instruction for all different $\theta$. Therefore, the generated images either have too many or too few changes. On the contrary, for InstructEdit there is no such mask threshold and the generated masks can exactly localize the intended region to be edited. By improving the mask quality, InstructEdit can faithfully do the edit and outperform DiffEdit. We show that without the

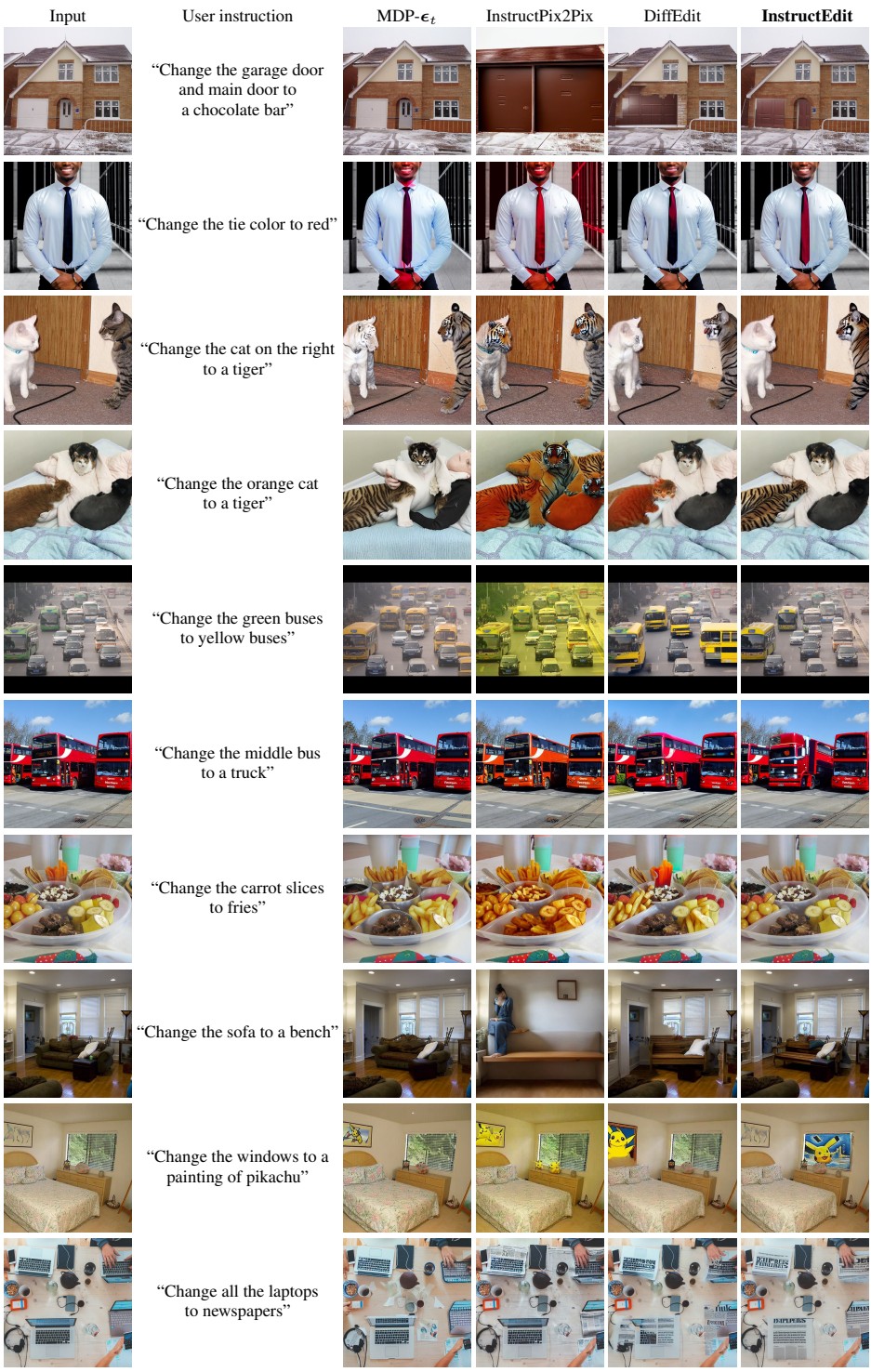

Figure 6: Qualitative results of baselines and our method. Here we use the same form of instruction "Change ... to ..." as an example.

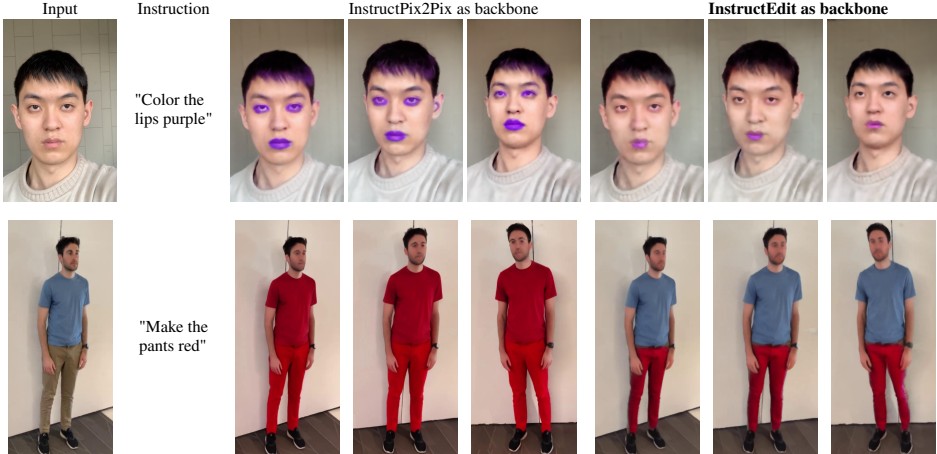

Figure 7: Comparison of the NeRF editing results of our method and InstructPix2Pix as backbone using Instruct-NeRF2NeRF as the editing pipeline.

help of the grounding pre-trained network, it is very hard for the diffusion model itself to accurately outline the region and do fine-grained editing at this scale.

### 4.4 INSTRUCTION PROCESSING

In Fig. 4 we show the results of an ablation study and how BLIP2 helps to improve the quality of the edited images. When the instruction does not clearly refer to all the prominent objects in the image, e.g. not mentioning a visible "dog" like in the the first example, the input and edited captions provided by ChatGPT do not contain "dog" as well. This leads to difficulties when the encoding ratio $r$ is increasing. In the second example, where an unclear "it" is used to refer to the object to be edited, without BLIP2 ChatGPT fails to correctly generate a segmentation prompt and the captions. In such a case, BLIP2 provides an extra description of the image such that ChatGPT can understand which object should be edited and provides improved captions.

We also show in Fig. 5 that InstructEdit is very robust and can correctly understand differently phrased instructions as input. This demonstrates the benefit of adopting a large language model to process the instruction rather than hard-coded parsing.

### 4.5 NeRF EDITING

We show that our framework can be used as an image editing backbone for a NeRF editing pipeline. We adopt Instruct-NeRF2NeRF (Haque et al., 2023), which is a NeRF editing method based on user instructions. Instruct-NeRF2NeRF iteratively updates images from the training dataset using InstructPix2Pix while optimizing the NeRF scene. Instead of using InstructPix2Pix as the image editor, we replace it with our method and follow the same editing pipeline as Instruct-NeRF2NeRF. We show the comparison in Figure 7 and accompanying video in the project webpage. Our method can do the edit within the desired region, while InstructPix2Pix tends to overedit the image.

## 5 DISCUSSION AND CONCLUSIONS

There are several limitations in our work. As ChatGPT and BLIP2 are probabilistic models, their outputs are not always optimal and thus may fail to correctly parse the user instruction. Also, as the editing is performed within the generated mask, which has the same shape as the object, it is difficult to conduct deformations. In this work, we proposed a framework termed InstructEdit that can do fine-grained editing and directly accept user instructions as input. We use a language processor to process user instructions, a segmenter to generate high-quality masks, and an image editor to do the mask-guided editing. Experiments show that our method outperforms previous editing methods when doing fine-grained edits. We improve the mask quality over DiffEdit and thus improve the quality of edited images. Our framework can be also combined with NeRF editing pipeline.

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

## A  REPRODUCIBILITY

Our implementation can be found in https://anonymous.4open.science/r/InstructEdit_Anonymous-D2CC.

## B  DIFFEDIT

DiffEdit is a mask-based text-guided image editing model that can automatically generate masks. During the denoising process, different text prompts will guide the diffusion model to yield different predictions. By contrasting two predictions, the difference can give information for which image regions the input caption and edited caption have different estimates. We refer to the text prompt that is used to generate or inverse the input image as input caption $\mathbf{c}^{(i)}$, and to the text prompt that is used to describe the edited image as edited caption $\mathbf{c}^{(e)}$. The predicted noise at each timestep $t$ guided by $\mathbf{c}^{(i)}$ and $\mathbf{c}^{(e)}$ are $\boldsymbol{\epsilon}_t^{(i)} = \boldsymbol{\epsilon}_\theta(\mathbf{x}_t, \mathbf{c}^{(i)}, t)$ and $\boldsymbol{\epsilon}_t^{(e)} = \boldsymbol{\epsilon}_\theta(\mathbf{x}_t, \mathbf{c}^{(e)}, t)$, respectively. The difference of the prediction $\boldsymbol{\epsilon}_t^{(d)}$ at timestep $t$ is then calculated as:

$$\boldsymbol{\epsilon}_t^{(d)} = \left| \boldsymbol{\epsilon}_t^{(i)} - \boldsymbol{\epsilon}_t^{(e)} \right|, \tag{5}$$

$\widetilde{\boldsymbol{\epsilon}}_t^{(d)}$ is finally decoded from the latent to a binary mask image $\mathbf{M}$ with a threshold $\theta$.

DiffEdit uses DDIM inversion to invert the input image $\mathbf{x}_0$ into the initial noise $\mathbf{x}_T$ which can generate $\mathbf{x}_0$. Each inversion step is calculated as:

$$\mathbf{x}_{t+1} = \sqrt{\alpha_{t+1}} \cdot f_\theta(\mathbf{x}_t, \mathbf{c}, t) + \sqrt{1 - \alpha_{t+1}} \cdot \boldsymbol{\epsilon}_\theta(\mathbf{x}_t, \mathbf{c}, t). \tag{6}$$

We iteratively apply this formula until we obtain $\mathbf{x}_T$. However, if we stop the inversion step at timestep $r \leq T$, we encode $\mathbf{x}_0$ into a less noised version $\mathbf{x}_r$. $r$ is called the encoding ratio as in DiffEdit. A larger value of $r$ indicates a stronger edit effect, making the edited image guided more by the edited caption but less similar to the input image.

## C  EDITING TYPES

In this work, we focus on image edits restricted to certain object categories or the number of occurrences in the input image. We consider replacing objects with other objects and changing the attributes of objects. The edited image should faithfully follow the user instructions while also preserving the layout and the appearance of the other parts in the original image.

- Single-object: there is a foreground object consisting of multiple parts and we want to edit one part of that object.
- Multi-object of the same type: there are several objects of the same type and we want to edit one or more of them.
- Multi-object of different types: there are several objects of different types and we want to edit one of them.

## D  IMPLEMENTATION DETAILS

### D.1  EXAMPLE OF TASK TEMPLATE

We show one task template we provide to ChatGPT:

*For example, if the user says "Change the dog to a cat", you need to give the segmentation model only the keyword "Dog". You also need to give the image editing model two text prompts: "Photo of a dog", and "Photo of a cat". Your answer should be in the form of: Segmentation prompt: Dog. Editing prompt 1: "Photo of a dog". Editing prompt2: "Photo of a cat".*

Here, *Editing prompt 1* is the input caption $\mathbf{c}^{(i)}$ and *Editing prompt 2* is the edited caption $\mathbf{c}^{(e)}$.

## D.2 BASELINES

For MDP-$\epsilon_t$, we fix the interpolation factor as 1 as default and vary the editing starting timestep and ending timestep; For InstructPix2Pix, we only tune the classifier-free guidance factor for text condition; For DiffEdit and InstructEdit, as they both share the same generation process after obtaining a mask, we tune the encoding ratio $r$. For DiffEdit we also tune the additional parameter $\theta$ which controls the threshold when computing a binary mask. For InstructPix2Pix and InstructEdit, we use the same user instruction as input, while for MDP-$\epsilon_t$ and DiffEdit we manually design the input caption and edited caption based on the user instruction.

For MDP-$\epsilon_t$, we use the official implementation from https://github.com/QianWangX/MDP-Diffusion. For InstructPix2Pix, we use the official implementation from https://github.com/timothybrooks/instruct-pix2pix. For DiffEdit, as there is no official implementation available, we refer to https://github.com/johnrobinsn/diffusion_experiments/blob/main/DiffEdit.ipynb and modify the parts that are not consistent with the paper (Couairon et al., 2022).

## D.3 NERF EDITING

We use Instruct-NeRF2NeRF as our editing pipeline. The original Instruct-NeRF2NeRF uses InstructPix2Pix as the image editing backbone. Here we replace it with InstructEdit. We refer to the implementation from https://github.com/ayaanzhaque/instruct-nerf2nerf which is developed based on the Nerfstudio (Tancik et al., 2023) framework. For the NeRF reconstruction stage, we use the default method for static scenes in Nerfstudio called *nerfacto* for 30000 iterations. For the NeRF editing stage, we use the *in2n* config for the InstructPix2Pix backbone but the *in2n_tiny* config for the InstructEdit backbone due to memory constraints caused by pre-trained models. Please refer to the documentation of Nerfstudio for the settings of these two configs. For both methods, we perform dataset updating for around 3000 iterations.

## E   MORE RESULTS

### E.1   QUALITATIVE AND QUANTITATIVE COMPARISON

We show more qualitative results in Figs. 13 and 14. We show that InstructEdit performs better than the baseline methods in the majority of cases.

For the user study, we could acquire 26 workers on MTurk who were presented with triplets of images, each triplet consisting of one input image and two edited images. One of the edited images always was the output from our method, the second one was a randomly picked edited image from one of the baseline methods. We also provide an additional user study for the 20 examples shown in Figs. 13 and 14. We presented triplets of images as done in the main paper, each triplet consisting of one input image and two edited images. One of the edited images was the output from our method, while the second one was a randomly picked edited image from one of the baseline methods. For every example, we ask the participant a question: *"Which image applied the instruction more appropriately?"*. Results show that InstructEdit was preferred over MDP-$\epsilon_t$, InstructPix2Pix, and DiffEdit in 67.5%, 61.0%, and 57.0% of the cases, respectively. We show a screenshot of the user study interface in Fig. 12.

### E.2   MASK IMPROVEMENT

We show more results for mask improvement of InstructEdit over DiffEdit in Fig. 15. The quality of the automatic masks generated by DiffEdit is highly affected by the mask threshold $\theta$. Nevertheless, the mask area generated by DiffEdit can be completely off the region that should be edited under different $\theta$. With the help of the grounding model and segmentation model, however, the generated mask can accurately outline the region to be edited without tuning $\theta$.

### E.3 ENCODING RATIO

We show the effect of increasing the encoding ratio and compare the results with an inpainting baseline in Fig. 8. The inpainting baseline from [2] is also an image editing framework, which also adopts the Grounded Segment Anything to extract a high-quality mask for the input image, but uses the Stable Diffusion Inpainting model [3] as the image editing model.

In general, increasing the encoding ratio can lead to a larger change in the edited image compared to the original image. The edited image will follow the user instruction more and look less like the input image. Compared to using an inpainting model as an image editing model, using the mask-guided generation enables more flexibility in choosing either a larger change or a smaller change. An inpainting model however will completely ignore the original pixel information inside the mask region and only follow the inpainting prompt.

## F  FAILURE CASES

We identify one common failure case for each component in our framework and show those cases in Figs. 9 to 11. For the language processor, BLIP2 sometimes may provide a description that is not helpful to the editing task. For example in Fig. 9, describing the brand of the car does not help change the color of the target car. For the segmenter, Grounded DINO could fail to correctly locate the object given a certain direction. In Fig. 10, the object to be edited should be the dog on the right, but Grounded DINO gives a larger probability score to the dog on the left and therefore segmenter produces the wrong mask. For the image editor, as all the editing is performed within the mask, the editor is not good at deforming the object as shown in Fig. 11.

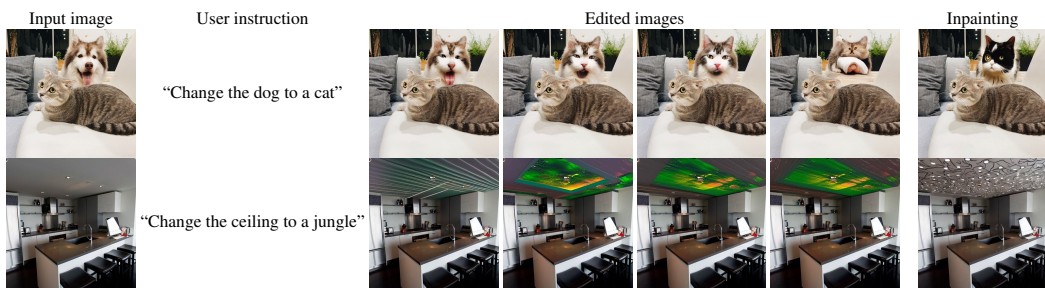

Figure 8: Effect of increasing encoding ratio and comparison with an inpainting baseline. From left to right the edited images are edited by increasing encoding ratios. Note that the inpainting baseline does not accept user instruction as input, but only inpainting prompt.

---

[2]https://github.com/IDEA-Research/Grounded-Segment-Anything/tree/main
[3]https://huggingface.co/runwayml/stable-diffusion-inpainting

Input image     Instruction & description     Edited image

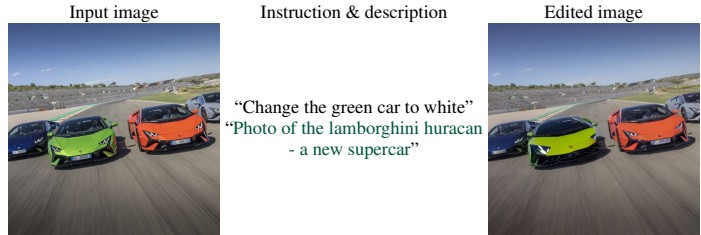

Figure 9: Failure case of the language processor. Note that we show the BLIP2 description below the user instruction. The generated BLIP2 description is irrelevant to the editing task.

Input image     Instruction & description     Generated mask

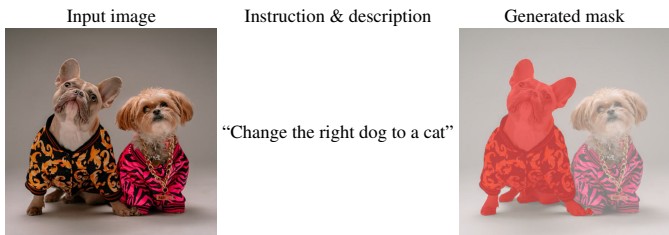

Figure 10: Failure case of the segmenter. The segmenter here fails to mask the dog on the right side.

Input image     Instruction & description     Edited image

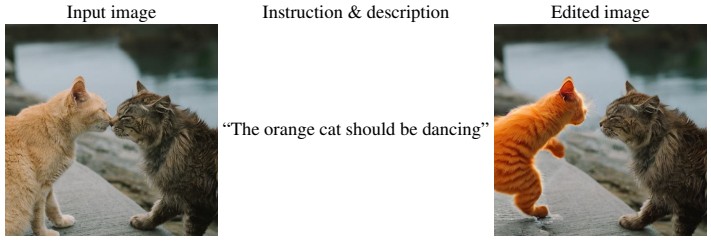

Figure 11: Failure case of the image editor. The edited cat fails to preserve the identity of the cat in the original image.

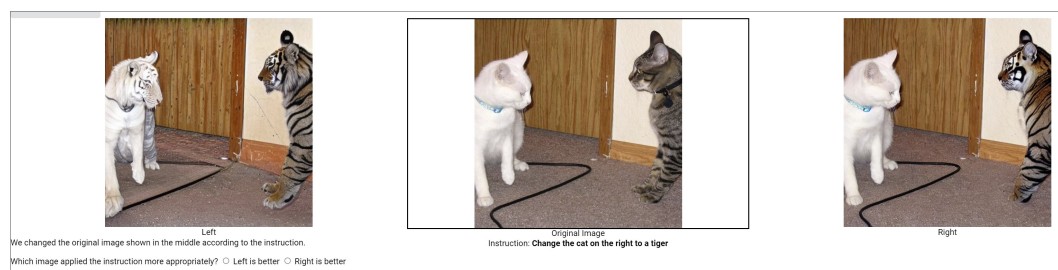

Figure 12: Screenshot of the user study interface.

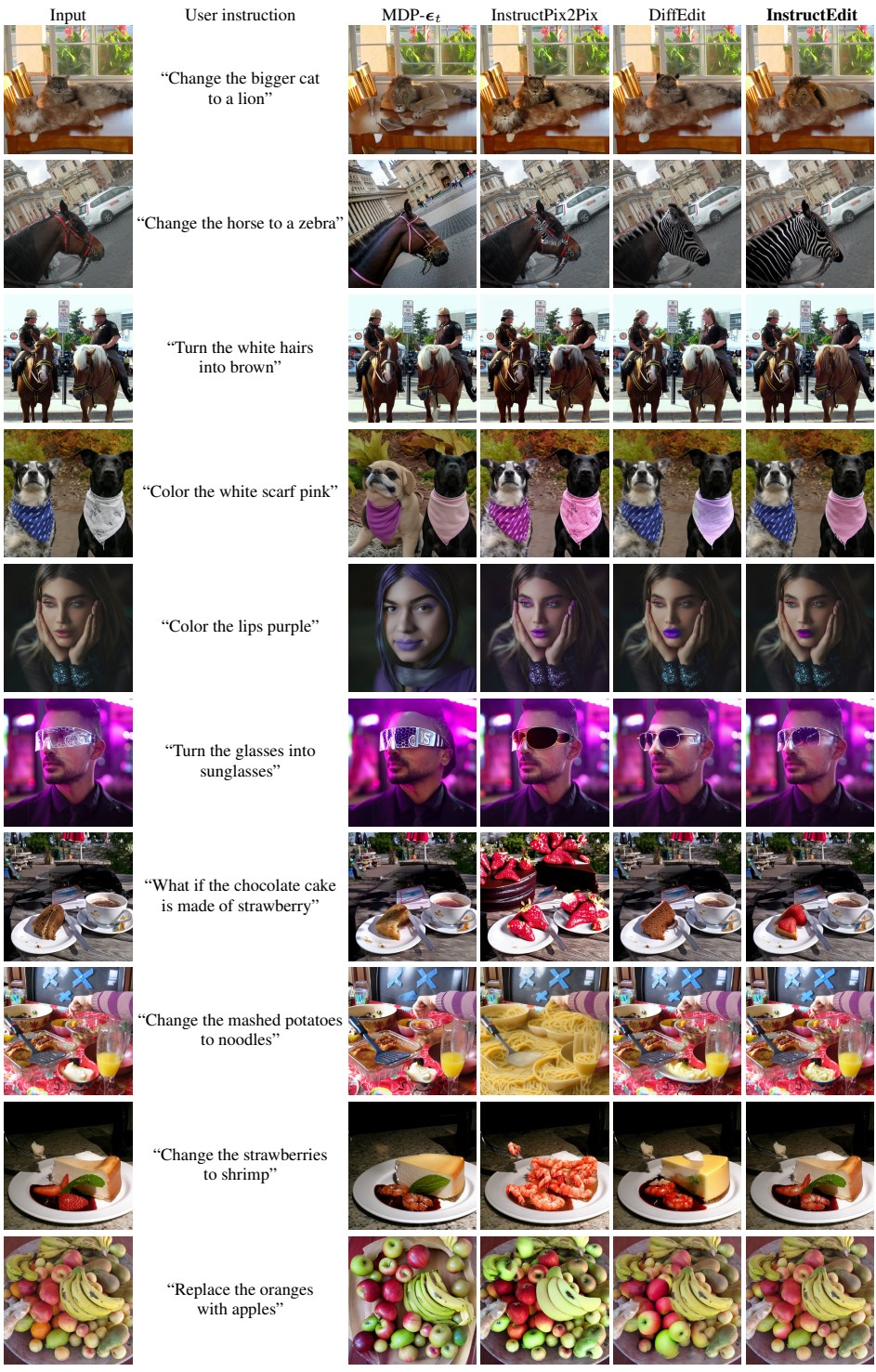

Figure 13: Qualitative results of baselines and our method.

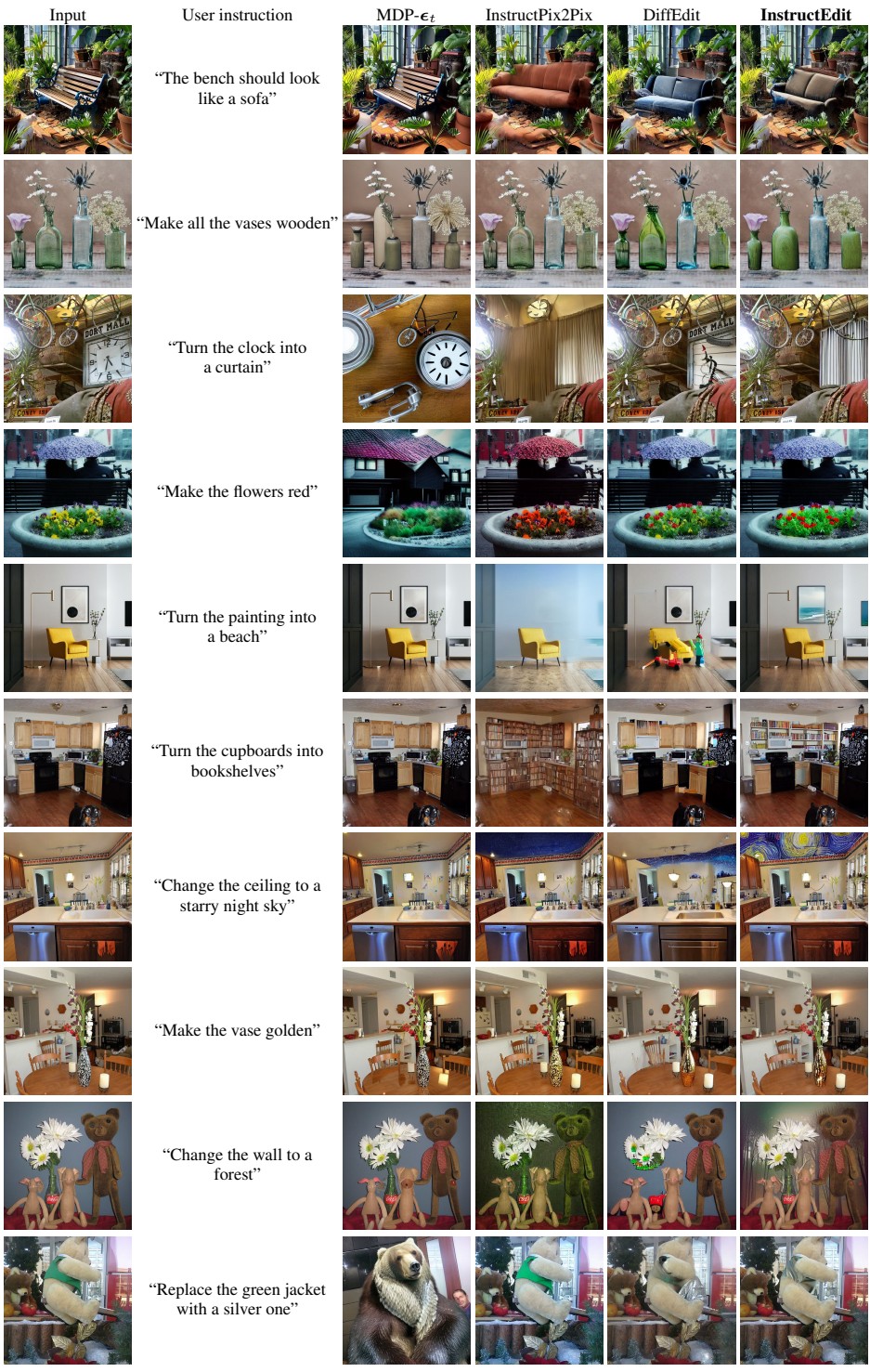

Figure 14: Qualitative results of baselines and our method.

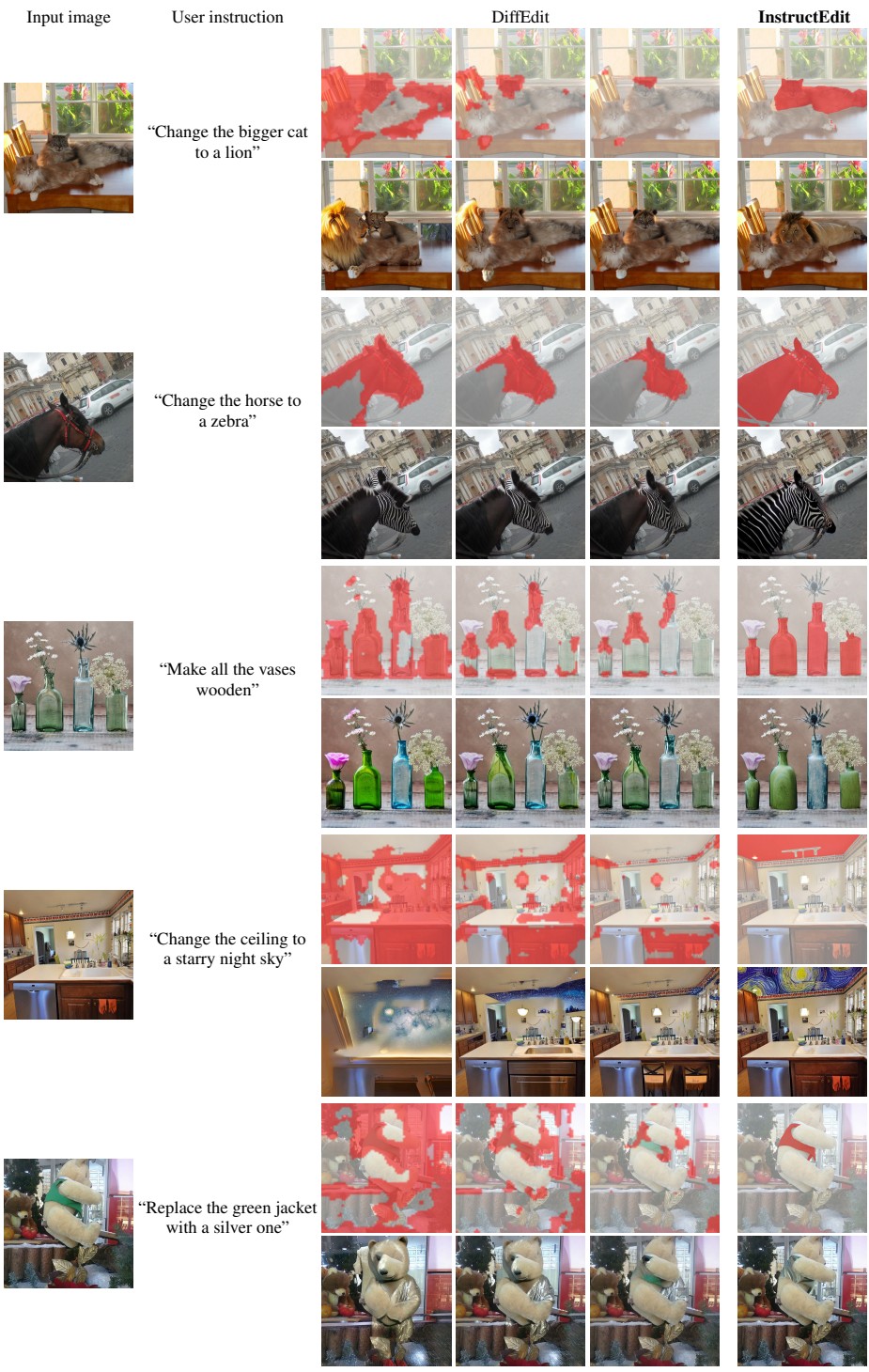

Figure 15: Comparison of the masks (colored in red and blended with the input image) and the corresponding edited image (below each mask) generated by DiffEdit and InstructEdit.

