# OpenReview forum: "InstructEdit: Improving Automatic Masks for Diffusion-based Image Editing With User Instructions"
_ICLR.cc/2024/Conference — ICLR 2024 Conference Withdrawn Submission_

### Official Review · Reviewer_YLK1 · 2023-10-13

**Soundness:** 2 fair
**Presentation:** 2 fair
**Contribution:** 1 poor
**Rating:** 3
**Confidence:** 4

**Summary:**

This paper presents InstructEdit, a method for fine-grained image editing using diffusion models. To this end, the authors propose a framework with three main components: a Language Processor (ChatGPT, BLIP2), a Segmenter (Grounded Segment Anything), and an Image Editor (DiffEdit). Experiments show that InstructEdit outperforms baselines on fine-grained editing tasks (e.g., editing images with multiple objects).

**Strengths:**

- Fine-grained image editing is an emerging topic, as I believe that it will allow more human control over the image editing process.
- The idea of using LLMs to refine humans' coarse instructions has also received attention from the community recently.
- The paper is generally easy to follow. The authors also did a good job providing an overview of related work.

**Weaknesses:**

My biggest concern about this paper are: (1) its novelty, and (2) it might not be a suitable submission for ICLR.

I feel that it is solely a system design aimed at improving DiffEdit (ICLR 2023). All proposed modules were directly adopted from other works without any technical contribution. My detailed justifications are as follows:

- Prior work (e.g., InstructPix2Pix (CVPR 2023) has introduced a framework that can perform edits given a written instruction (not input caption and edited caption), so it might be inappropriate to claim this as a major contribution in novelty aspect.
- Improvement of mask-quality directly comes from pretrained Grounded Segment Anything.

Given that (1) This paper focuses solely on image editing applications, and (2) There is actually no "representation learning" happening.
I believe that ICLR is not a suitable venue for this submission. Perhaps CVPR, ICCV, ECCV, or SIGGRAPH might be more suitable.
Thus, my initial recommendation is to reject, authors please submit this work to CVPR.

*Misc:*
- I am confused by 'multiple forms of user instructions as input.' As far as I understand, InstructEdit can only take input in text form (e.g., 'Change it to a dog'). What do the authors mean by 'multiple forms' (e.g., segmentation mask, reference image)?

- Quantitative data on only 10 images seems too small. While I am not suggesting using > 1k images, I think for reliable quantitative results, it should be around 100 images (in line with Imagic (CVPR 2023), Null-text Inversion (CVPR 2023))

- There is a typo in the Abstract on OpenReview: "Abstract: [...] We adopt Stable Diffusion and the mask-guided generation from DiffEdit for this purpose. *What are the main achievements, outcomes, results?* Experiments show that our method outperforms previous editing methods in fine-grained editing applications where the input image contains a complex object or multiple objects."

- Figure 5 can be improved. While I am quite impressed by the left side (Room photo), the right side (girl photo) seems too short. Authors can consider making this figure into 2 rows and providing more edits for the girl photo.

**Questions:**

- Is there any way to validate cumulative errors in the three components?
InstructEdit relies on many pretrained models, and each can produce errors (as I see in the Supplementary). Is there any way to assess the cumulative errors of all components?

- What happens if ChatGPT gives two segmentation prompt objects?
I have played with ChatGPT, using the same template provided by the authors in the Supplementary (D1), and found cases where ChatGPT will answer that two objects need to be segmented (instead of only one). Have the authors faced the same situation?

My prompt: "Change the dog next to the cat to a lion".
ChatGPT answer:
"Segmentation prompts: Dog, Cat.
Editing prompt 1: "Photo of a dog next to a cat".
Editing prompt 2: "Photo of a lion next to a cat".

**Details Of Ethics Concerns:**

No ethics concerns.

---

### Official Review · Reviewer_6zrz · 2023-10-30

**Soundness:** 2 fair
**Presentation:** 3 good
**Contribution:** 2 fair
**Rating:** 3
**Confidence:** 4

**Summary:**

The authors proposed to apply a cascade of models to achieve instruction-based image editing. The pipeline includes an LLM for instruction processing, a segmentation model to obtain mask, and a mask based image editor for image editing.

**Strengths:**

- A well-designed engineer pipeline for instruction-based editing, which seems good for some practical application and outperforms other similar methods.
- A good comparison among different pipelines.

**Weaknesses:**

- It's unfair to claim the model as a general instruction-based editing, since the proposed method may be only working for object-level editing. For editing with stronger identity preservation like 'make the face brighter, change the grass into green, make the season to winter' etc, the proposed method is not expected to work well.
- The proposed pipeline is a good engineer pipeline, but it's not clear that for engineer-based pipeline, why the current method is better than a workflow like: user used segmentation methods to indicate the editing regions and run inpainting like photoshop generative fill.
- InstructPix2Pix is not designed for real images, so the comparison is not fair enough.
- The proposed pipeline is very hard to deploy as a single pipeline, so lacking enough scientific values for researchers or industrial leads.

**Questions:**

See above.

---

### Official Review · Reviewer_pqYf · 2023-10-31

**Soundness:** 3 good
**Presentation:** 3 good
**Contribution:** 2 fair
**Rating:** 3
**Confidence:** 4

**Summary:**

InstructEdit, the system introduced in this paper, enables text-guided image editing. Leveraging a language model (ChatGPT) and, optionally, an image captioner (BLIP2), it extracts the object in question and rewrites the instruction into an edited caption. Subsequently, it employs the Grounded Segment Anything model to segment the intended region. Finally, the mask and edited caption are inputted into DiffEdit to produce the edited image. A small-scale qualitative user study (involving 26 reviewers and 20 examples) indicates a preference for InstructEdit over competing methods (MDP-$\epsilon_t$, Instruct-Pix2Pix, and DiffEdit) in 57.0% - 67.5% of cases. The paper further explores InstructEdit's applicability to NeRF model editing and acknowledges several of its limitations.

**Strengths:**

**Originality**: The paper introduces a novel integration of ChatGPT, BLIP2, Grounded SAM, and DiffEdit for enhanced text-guided image editing. Its originality is marked by this unique combination, although the approach itself may be considered somewhat straightforward.

**Quality**: The paper addresses the issue of ambiguous edit instructions through the use of masks, building on a foundation laid by prior research. The methodology is technically sound, showcasing notable advancements over standard methods, particularly in images featuring complex or multiple objects. However, the scope of the experiments could benefit from expansion (details in the weaknesses section).

**Clarity**: The paper is well-organized and generally comprehensible. Nonetheless, certain figures might cause confusion and would benefit from further clarification (details in the questions section).

**Significance**: The paper's most significant limitation lies in its impact. The integration of existing models, while practical, offers limited theoretical or empirical advancement, leading to a potentially modest contribution to the field.

**Weaknesses:**

The paper lacks a comprehensive theoretical and empirical analysis of its proposed system. Critically:

1. Reliance on pre-built models may hinder further innovation. Although the paper acknowledges certain limitations, such as suboptimal outputs from the ChatGPT + BLIP2 combination, it offers no clear strategy for remediation. This lack of guidance leaves users without a roadmap for improvement when issues arise.

2. The individual contributions of each component to the overall system are unclear. Questions like the accuracy of ChatGPT in extracting the object of interest, and how this impacts the final image edit quality, need addressing. At the bare minimum, the paper should build a test set to quantitatively evaluate this (even with subjective measures). Moreover, exploring alternative component configurations, like substituting ChatGPT with Chat LLaMa 2, could show potential trade-offs in accuracy and response time.

3. "Blended Latent Diffusion" is a relevant methodology that could be worth comparison. It could also potentially serve as an alternative to the DiffEdit component in the proposed framework.

4. The system's overall latency is unclear. It may be significantly slower than approaches like Instruct-Pix2Pix, raising questions about the justification for potential trade-offs between speed and accuracy.

5. Incorporating a segmenter may inadvertently constrain the system's editing capabilities, especially for commands like "add a hat" or "turn into Van Gogh style." Understanding these limitations is crucial for assessing the system's versatility.

**Questions:**

In addition to the previously mentioned concerns, the presentation raises several minor issues:

6. Figure 1, the subplot for "Change the buildings to palaces" -- it exhibits noticeable artifacts in the sky. This suggests potential deficiencies in either the masking or blending processes. Clarification on the cause of these artifacts and proposed solutions would enhance the paper's rigor.

7. The examples presented in Figure 4 may not effectively showcase the strengths of BLIP2. The depicted fox appears relatively good at lower encoding ratios, while higher ratios introduce significant artifacts regardless of BLIP2's involvement. A more illustrative example might better demonstrate BLIP2's efficacy. This reinforces the need for a more precise quantification of each component's contribution to the system's overall performance.

---

### Official Review · Reviewer_PLFP · 2023-11-01

**Soundness:** 3 good
**Presentation:** 3 good
**Contribution:** 2 fair
**Rating:** 6
**Confidence:** 4

**Summary:**

This paper proposes InstructEdit, a framework for diffusion-based fine-grained editing with user instructions. InstructEdit uses large pre-trained models to edit images following user instructions based on DiffEdit. It accepts user instruction as input instead of input caption and edited caption for both the source and target images. InstructEdit has three components: a language processor adopting ChatGPT and optionally BLIP2 to parse the user instruction and output segmentation prompts and editing captions, a segmenter employing state-of-the-art Grounded Segment Anything to automatically generate a high-quality mask based on the segmentation prompt, and an image editor that uses the captions from the language processor and the masks from the segmenter to compute the edited image. The image editor adopts Stable Diffusion and the mask-guided generation from DiffEdit. Experiments show that InstructEdit outperforms baselines in fine-grained editing where the input image contains a complex object or multiple objects. it improves the
mask quality over DiffEdit and thus improves the quality of edited images. The framework can accept multiple forms of user instructions as input and be extended to NeRF editing.

**Strengths:**

- The paper is generally well-written and easy to follow. The symbols, terms, and concepts are adequately defined or clearly explained. The language usage is good.

- The proposed method is very simple and easy to understand. Sufficient details are provided for the readers.

- The experiments are well-executed. The empirical results and analyses are useful to show the advantages of the proposed method.

- The relevant literature is well-discussed and organized.

**Weaknesses:**

- The actual technical contribution of the paper is a bit slim. The paper seems just to improve the input mask quality over DiffEdit and thus improve the quality of edited images, which may not be considered a very significant contribution. Also, the improved technique, Grounded SAM, is not the contribution of the paper. Several previous techniques are combined. Still, the idea of involving a large language model for caption generation based on user instruction and adding multi-modal inputs is interesting. Please consider mentioning DALL·E 3.

- Despite the promising results, the proposed InstructEdit still has certain failure cases and limitations, such as changing the color of the buses. Providing more discussions on this could strengthen this paper further.

-  Most of the results presented in the paper only involve some simple cases, such as the object appearance/color change. The reviewer is curious if InstructEdit can be applied to more challenging scenarios involving large-geometry change, such as object position/size editing.

- The reviewer is interested in whether the good capability of the proposed approach can be applied to videos aside from the NeRF editing.

**Questions:**

- Will the authors release all the code, models, and data to ensure the reproducibility of this work?